# Differences in a Single Extracellular Residue Underlie Adhesive Functions of Two Zebrafish Aqp0s

**DOI:** 10.3390/cells10082005

**Published:** 2021-08-06

**Authors:** Irene Vorontsova, James E. Hall, Thomas F. Schilling, Noriaki Nagai, Yosuke Nakazawa

**Affiliations:** 1Physiology and Biophysics, University of California, Irvine, CA 92697, USA; ivoronts@uci.edu (I.V.); jhall@uci.edu (J.E.H.); 2Developmental and Cell Biology, University of California, Irvine, CA 92697, USA; tschilli@uci.edu; 3Neurobiology and Behavior, University of California, Irvine, CA 92697, USA; 4Faculty of Pharmacy, Kindai University, Osaka 577-8502, Japan; nagai_n@phar.kindai.ac.jp; 5School of Pharmaceutical Sciences, Keio University, Tokyo 105-8512, Japan

**Keywords:** aquaporin 0, AQP0, membrane intrinsic protein, MIP, adhesion, zebrafish, ocular lens, gene duplication

## Abstract

Aquaporin 0 (AQP0) is the most abundant lens membrane protein, and loss of function in human and animal models leads to cataract formation. AQP0 has several functions in the lens including water transport and adhesion. Since lens optics rely on strict tissue architecture achieved by compact cell-to-cell adhesion between lens fiber cells, understanding how AQP0 contributes to adhesion would shed light on normal lens physiology and pathophysiology. We show in an in vitro adhesion assay that one of two closely related zebrafish Aqp0s, Aqp0b, has strong auto-adhesive properties while Aqp0a does not. The difference appears to be largely due to a single amino acid difference at residue 110 in the extracellular C-loop, which is T in Aqp0a and N in Aqp0b. Similarly, P110 is the key residue required for adhesion in mammalian AQP0, highlighting the importance of residue 110 in AQP0 cell-to-cell adhesion in vertebrate lenses as well as the divergence of adhesive and water permeability functions in zebrafish duplicates.

## 1. Introduction

Membrane intrinsic protein (MIP), also known as aquaporin 0 (AQP0), is the most abundant lens membrane protein. Absence or mutation thereof leads to cataract formation [1]. It has been shown to have a number of functions in the lens including water transport, adhesion, and regulation of functions of other proteins, such as connexin 50 [2] and filensin [3,4]. Historically, water transport has been the most studied function of AQP0, largely in expression systems, but in recent years, adhesive properties of AQP0 have begun to receive more attention. Since lens optics depend on the precise architecture achieved by compact cell-to-cell adhesion, understanding how proper cell-to-cell adhesion is achieved would shed light on normal lens physiology and pathophysiology.

A structural role for AQP0 was first proposed based on observations of square array junctions [5] and later confirmed by others [6,7]. Self-adhesive properties of human AQP0 were first demonstrated in vitro by expressing AQP0 in adhesion-deficient mouse fibroblast L cells [8]. This assay utilized the L cells stably expressing AQP0, dyed red and seeded in a dish, on the top of which AQP0-expressing cells dyed blue were loaded. The percentage of adherent cells was then quantified by flow cytometry. A human AQP0 mutation, R33C, had reduced adhesive properties, while water permeability was unaffected [9]. Bovine AQP0 reconstituted into large unilamellar liposomes increased adhesion via electrostatic interactions with the negatively charged membrane [10]. Similarly, mouse AQP0 was shown to adhere to untransfected adhesion-deficient mouse fibroblast L cells, negatively charged L-α-phosphatidylserine large unilamellar vesicles, and lens vesicles derived from the WT or AQP0-KO mouse lens cortex [11]. Electron crystallography of the sheep lens suggested that AQP0 connects through ^109^Pro, ^110^Pro, ^113^Arg, and ^123^Pro in the C-loop between two AQP0 molecules on adjoining membranes [12]. By mutating these residues in rat AQP0 and transfecting them in vitro, we confirmed their importance utilizing the aforementioned adhesion assay [13]. P109A/P110A significantly reduced adhesion between homotypic pairs of cells, while AQP0 V107I and R113G mutations did not alter adhesion [13].

In mammalian systems, however, it has been impossible to study adhesion separately from other functions of AQP0. For this reason, it is useful to study other models with duplicated AQP0 orthologs, including teleost fishes such as zebrafish, medaka, and Atlantic salmon. Tetraploid Atlantic salmon (*Salmo salar*) AQP0 channels (*Aqp0a1, 0a2, 0b1, 0b2*) are all highly expressed in the lens and all permeate water in the *Xenopus laevis* oocyte expression system with differing pH sensitivities [14]. All four Aqp0s have strong auto-adhesive properties and lower heterotypic adhesive affinities to other salmon Aqp0 paralogs as well as human AQP0 [15]. This suggests that the adhesive properties of Aqp0s in teleosts are likely conserved with their mammalian orthologs, making them useful for studying the underlying mechanisms. Here, we utilized the zebrafish, which we had shown partitions at least some of the functions of AQP0 among the two duplicated genes, *aqp0a* and *aqp0b* [16,17,18], but whose adhesive properties had not been tested.

Aqp0b-mediated water permeability appears to be unnecessary for adult lens transparency [18], but Aqp0b is likely to have an additional function, possibly adhesion. While both Aqp0a and Aqp0b permeate water in the *Xenopus laevis* oocyte expression system [14], Aqp0b lacking the water transport function can rescue the cataract formed in 3-day-old zebrafish lenses after Aqp0b knock-down [16]. An adhesive function for Aqp0b may also help explain why a non-water-permeable variant of Aqp0b (S19 compared to G19) has also been identified in some strains of zebrafish [14,16,17,19]. In this study, we tested the adhesive properties of zebrafish Aqp0a and Aqp0b in homotypic and heterotypic stably expressed L-type mouse fibroblast cell lines. In addition, we performed a mutational screen to identify key residues required for these functions.

## 2. Materials and Methods

### 2.1. Cell Culture

Mouse fibroblast L cells (ATCC cell bank, Manassas, VA, USA) were cultured under standard culture conditions of 5% CO_2_ at 37 °C in Dulbecco’s modified Eagle’s minimum essential medium (DMEM; Nacalai Tesque, Kyoto, Japan) with a penicillin–streptomycin antibiotic mixture (100 U/mL and 100 mg/mL, respectively) containing 10% fetal calf serum (FBS; Life Technologies, Waltham, CA, USA).

### 2.2. Cell-to-Cell Adhesion Assay

Cell-to-cell adhesion assays were carried out as previously described [13]. Briefly, stable cell lines were established expressing pcDNA-Aqp0a, pcDNA-Aqp0b, or pcDNA-MIPfun wildtype\ or mutant forms. WT pcDNA-Aqp0a and WT pcDNA-Aqp0b were previously generated in the Hall lab. MIPfun was cloned into pcDNA3.1 (+) using XbaI and HindIII restriction sites with CTGGCTAGCGTTTAAACTTAATGTGGGAGTTCAGGTCC (forward) and AGCGGGTTTAAACGGGCCCTTTATAGGGCCTGCGTCTTC (reverse) primers using a Gibson Assembly Cloning Kit (E5510S New England Biolabs Inc, Ipswich, MA, USA). Cells transfected with empty pcDNA3 were used as negative controls. Two groups with equal numbers of cells in each cell line were generated. The first group was loaded with CellTracker Red CMPTX dye (Life Technologies) and cultured for 18 h until confluent. The second group of cells was loaded with CellTracker Blue CMAC dye (Life Technologies) and seeded over the first layer for 18 h. After washing non-adherent cells off with PBS three times, the percentage of red or blue positive cells was measured by flow cytometry (Aria III, high-speed multicolor digital analyzer, BD Bioscience, Franklin Lakes, NJ, USA).

### 2.3. Statistical Analysis

One-way ANOVA with post-hoc Tukey’s multiple comparisons was used to test for statistically significant differences in adhesion between groups (SPSS version 24; IBM Corporation, Armonk, New York, NY, USA). Each experiment was performed at least six times.

### 2.4. Western Blot Analysis

Western blot analysis was carried out as previously described [13] with minor modifications. Briefly, cell lysates were prepared in the EBC lysis buffer (120 mM NaCl, 0.5% NP-40, 50 mM Tris HCl (pH 8.0)) with a protease inhibitor cocktail (Nacalai Tesque, Kyoto, Japan). Protein concentrations were measured using a Bradford assay. Equal amounts of protein samples were separated by SDS-PAGE and transferred to polyvinylidene difluoride (PVDF) membranes (Millipore, Burlington, MA, USA). To quantify protein expression levels, we tested a number of anti-mammalian AQP0 antibodies against zebrafish lens homogenates, and the polyclonal rabbit anti-human AQP0 C-terminus antibodies (H44, Santa Cruz sc-99059) gave the strongest and cleanest signal in detecting both, Aqp0a and Aqp0b, by Western blotting (data not shown). This antibody does, however, have a stronger affinity for rat AQP0 (total score of 81.3 and 84.09% identity) compared to zebrafish Aqp0s, which are similar—Aqp0a (total score of 62.4 and 61.35% identity) and Aqp0b (total score of 58.9 and 54.55% identity; Appendix A). Polyclonal antisera against zebrafish Aqp0a C-terminus and zebrafish Aqp0b C-terminus [17] shown to be ortholog-specific [18] were purified (Econo-Pac Serum IgG; Bio-Rad Laboratories, Hercules, CA, USA). Anti-β-actin antibody (C-4, Santa Cruz Biotechnologies, Santa Cruz, CA, USA) was used as a loading control. Anti-rabbit HRP-conjugated secondary antibodies were used (Cell Signaling Technology, Danvers, MA, USA). Image J was used to quantify protein loading in at least three different Western blots.

### 2.5. Mutagenesis

Protein sequence alignments were carried out using an NCBI Constraint-based Multiple Alignment Tool (COBALT) and viewed using the NCBI Multiple Sequence Alignment Viewer, version 1.18.1. Protein sequence IDs for *Homo sapiens* AQP0—NP_036196.1; for *Bos taurus* AQP0—NP_776362.1; for *Mus musculus*—NP_032626.2; for *Rattus norvegicus*—NP_001099189.1; for *Fundulus heteroclitus* MIPfun—AAF04146.1; for *Danio rerio* Aqp0a and Aqp0b— NP_001003534.1 and NP_001018356.1; for *Salmo salar* Aqp0a1, Aqp0a2, Aqp0b1, and Aqp0b2—AJD87691.1, AJD87688.1, AJD87689.1, and AJD87690.1, respectively. Mutations were generated using a QuikChange II XL site-directed mutagenesis kit (200521 Agilent, Santa Clara, CA, USA) and confirmed by sequencing (Genewiz, South Plainfield, NJ, USA). Primers (forward, reverse) used to generate Aqp0a T110N were 5′-cagatttcctctcatattgtttggcgtgaccccataaag-3′ and 5′-ctttatggggtcacgccaaacaatatgagaggaaatctg-3′); Aqp0a Y195T—5′-agtggttaatgaagttcctagtgagcacagcaggggcaaaa-3′ and 5′-ttttgcccctgctgtgctcactaggaacttcattaaccact-3′; Aqp0b N110T or MIPfun N110T—5′-gtgcccctcatgttggtgggtgtaactccgt-3′ and 5′-acggagttacacccaccaacatgaggggcac-3′; Aqp0b T195Y—5′-cagtggttgatgaaattcctatagataacagcaggggcgaaagac-3′ and 5′-gtctttcgcccctgctgttatctataggaatttcatcaaccactg-3′; MIPfun N110P—5′-aggttccccctcatgttgggaggtgtgactccatacag-3′ and 5′-ctgtatggagtcacacctcccaacatgagggggaacct-3′.

## 3. Results

### 3.1. Zebrafish Aqp0b Has Adhesive Properties

We first tested the ability of WT zebrafish Aqp0a and Aqp0b to adhere to themselves when stably expressed in L-type mouse fibroblast cell lines. Adhesion was quantified using flow cytometry (Appendix A). Compared to the positive controls expressing rat AQP0 (rAQP0), with which ~43% of cells adhered, only ~18% of zebrafish Aqp0a-expressing cells adhered. This was similar to negative controls, either cells expressing rat AQP1 (rAQP1, ~10%) or empty vector transfected cells (L-(−); ~10%) (Figure 1A). In contrast, ~40% of Aqp0b-expressing cells were adherent, similar to rat AQP0. Although Western blot analysis (Figure 1B) showed similar protein expression levels for rat and zebrafish Aqp0s, we can only reliably compare levels between zebrafish Aqp0a and Aqp0b because they have similar predicted antibody affinity (Appendix A).

When expressed heterotypically, Aqp0b-expressing cells seeded on top of Aqp0a-expressing cells showed greater adhesion compared to homotypic Aqp0a cell pairs (~26% versus ~18%) but much weaker adhesion than homotypic Aqp0b-expressing cells (~39%) (Figure 1C). Aqp0a-expressing cells seeded on top of Aqp0b adhered poorly (~22%). In addition, the S19 Aqp0b variant adhered slightly less well than the G19 variant (Appendix A).

### 3.2. Residues 110 and 195 Differ between Aqp0a and Aqp0b

To pinpoint the residues responsible for the difference in adhesive properties between zebrafish Aqp0a and Aqp0b, the A-, C-, and E-loop amino acids were aligned with mammalian AQP0 and Atlantic salmon Aqp0s (Figure 2). Compared with human [8], rat [13] and salmon orthologs [15], the key residues differing in zebrafish Aqp0a were T110 and Y195. Aqp0b has N110 similar to salmon Aqp0s and T195 similar to mammalian AQP0 proteins.

### 3.3. N110 Is Key in Aqp0b-Mediated Adhesion

To test if either of these two residues in zebrafish Aqp0s affected adhesive properties, point mutations were induced in constructs of Aqp0a and Aqp0b and stably expressed in L-type mouse fibroblast cells (Figure 3A). Homotypic expression showed that N110T mutants significantly increased adhesion of Aqp0a from ~18% in WT to ~30%, while Y195T mutants were not significantly different (~18% to ~23%). The double mutant Aqp0a T110N/Y/195T had increased adhesive properties of the same order as the N110T mutant alone at ~32%. Aqp0b N110T had a significantly decreased adhesion from ~40% in WT to ~29%, and in the double mutant N110T/T195Y to ~24%, while T195Y was not significantly different from WT at ~33%. Protein expression levels were confirmed to be similar by Western blotting using anti-zebrafish Aqp0a and anti-zebrafish Aqp0b sera (Figure 3B,C).

### 3.4. MIPfun and Aqp0b Have Similar Adhesive Properties

We next tested adhesion of the *Fundulus heteroclitus* AQP0, MIPfun, which had previously been shown to rescue cataract phenotypes induced by knockdown of zebrafish Aqp0a or Aqp0b in embryonic fish [17]. MIPfun stably expressed in L-type mouse fibroblast cell lines exhibited homotypic adhesive properties (~30%) (Figure 4A). Like Aqp0b, MIPfun has N110, and an N110T mutation reduced its adhesive properties similarly to Aqp0b. We also tested if an N110P mutation mimicking the mammalian residue at this key site for adhesion affected its homotypic adhesion. Indeed, N110P mutants increased MIPfun’s adhesive properties to ~40%. Western blotting using anti-zebrafish Aqp0b sera was used to ensure equal protein expression in stable lines of WT and mutant MIPfun (Figure 4B,C). The peptide used to generate the anti-zebrafish Aqp0b sera [17] has 100% identity with MIPfun.

## 4. Discussion

We showed a functional difference in adhesive properties between zebrafish Aqp0a and Aqp0b in an in vitro adhesion assay. Aqp0b has strong auto-adhesive properties while Aqp0a does not. The difference appears to be largely due to an amino acid difference at residue 110 in the extracellular C-loop. Aqp0a has a T at position 110 while Aqp0b has an N. This difference highlights the conservation of residue 110 and its importance in adhesion between zebrafish Aqp0b and mammalian AQP0, for which P110 (as well as P109) is a key residue required for adhesion [13]. It also suggests that zebrafish Aqp0a has largely lost adhesive activity and that its contribution to lens transparency is largely due to its water permeability. These results clarify genetic studies in zebrafish showing that Aqp0a is essential for lens development and transparency while Aqp0b is not, at least under homeostatic conditions, providing some of the first evidence separating requirements for AQP0′s functions in water permeability versus adhesion [18].

We show that Aqp0b promotes adhesion when stably expressed in L-type mouse fibroblast cells while Aqp0a does not. While at face value rat AQP0 and zebrafish Aqp0b appear to have similar levels of adhesion in homotypic cell lines (Figure 1), we cannot make quantitative comparisons between absolute protein expression levels between species. We also cannot compare absolute adhesive properties between species either since the anti-human AQP0 antibody has a higher predicted affinity to rat AQP0. However, differences in the relative adhesive properties between Aqp0a and Aqp0b can be reliably compared since the predicted affinity of the antibody is similar for these two proteins.

Interestingly, the adhesion of Aqp0b plated on top of Aqp0a is slightly higher than of Aqp0a plated on Aqp0a, suggesting that Aqp0b can weakly adhere to Aqp0a or directly to the membrane. Previous studies using electron crystallography and adhesion assays have indicated the importance of extracellular loops for AQP0–AQP0 adhesion [12]. Protein alignments of the A-, C-, and E-loops reveal that residues T110 and Y195 differ between zebrafish Aqp0a and Aqp0b, as well as between Aqp0a and Aqp0s in other species with known adhesive properties (Figure 2). Swapping residues between Aqp0b and Aqp0a at sites 110 and 195 results in reduced adhesion in Aqp0b N110T and increased adhesion in Aqp0a T110N, while mutating residue 195 has no effect (Figure 3). Interestingly, these two mutations alone do not increase adhesion mediated by Aqp0a to similar levels as does WT Aqp0b, and similarly adhesion mediated by Aqp0b does not decrease to the same level as does Aqp0a with both mutations. These results confirm the importance of N110 and the extracellular C-loop in the adhesive properties of Aqp0b. Furthermore, it is interesting that adhesion mediated by zebrafish Aqp0a is so weak, while all the Atlantic salmon Aqp0s show strong adhesive properties.

In addition, we show that a similar N110T mutation in the Aqp0 of *Fundulus heteroclitus*, MIPfun, also reduces adhesion (Figure 4). This result is consistent with evidence that MIPfun has both water transport [17] and adhesive functions. Knockdown or loss-of-function mutants in zebrafish *aqp0a* or *aqp0b* increases the frequency of cataract at 2 and 3 days post-fertilization, which recovers by 4 dpf [16,17,18], and both can be rescued by transient overexpression of WT MIPfun. In contrast, a MIPfun N68Q mutant lacking water permeability rescues loss of Aqp0b but not of Aqp0a [16]. Thus, Aqp0a requires water transport function for embryonic lens transparency while Aqp0b does not. In the future, we will test the ability of MIPfun N110T with reduced adhesive properties to rescue an embryonic cataract in non-functional *aqp0b^−^*^/*−*^ mutant zebrafish. Although they form cataracts initially, *aqp0b^−^*^/−^ mutant lenses are indistinguishable from WT in terms of transparency, size, gross morphology, and optics after 4 dpf [18,20]. This suggests that the adhesion function of Aqp0b is dispensable once the lens matures, at least under homeostatic conditions, and can be compensated by other mechanisms. Other cell-to-cell adhesion mechanisms could potentially compensate for loss of adhesion by Aqp0b such as N-cadherin junctions [12] or connexin 50 [21]. Furthermore, *aqp0b^−^*^/*−*^ mutant lenses may also be sensitive to stress, e.g., osmotic, or mechanical perturbation. Future studies are needed to test the mechanical integrity and stiffness of *aqp0b^−^*^/*−*^ mutant lenses under load [22].

While there is high conservation between species in the extracellular loop residues of Aqp0s, there are variations between mammalian and piscine species in specific regions (Table 1). Notably, while P109 is conserved in all the species tested, P110 in mammals is mainly N110 in piscine species, as well as piscine N111 and M112, instead of mammalian A111 and V112. In the E-loop, site 195 varies from T195 in mammals to V195 and F195 in Atlantic salmon and killifish, while zebrafish Aqp0b has T195 and Aqp0a has Y195. It also appears that mammalian P110 yields a higher level of auto-adhesion compared to N110, as seen in MIPfun N110P (Figure 4). This suggests that mammalian AQP0 may have stronger cell-to-cell adhesion than piscine species, possibly to support mechanical integrity of the larger mammalian lens. Since sheep AQP0 is thought to connect through ^109^Pro, ^110^Pro, ^113^Arg, and ^123^Pro in the C-loop [12], it is likely that zebrafish Aqp0b may also connect to other Aqp0b through ^109^Pro, ^110^Asn, ^113^Arg, and ^123^Pro.

It is also interesting to speculate that having a proline with a nonpolar aliphatic residue that results in stronger auto-adhesion compared to an asparagine with polar neutral side chains leads to stronger adhesion as we showed for MIPfun. Molecular dynamics simulations could test this by quantifying the absolute adhesive force exerted per protein molecule to test how changes in N110P and other residues in the extracellular loops affect strength of AQP0–AQP0 interactions. If indeed mammalian AQP0 can exert stronger adhesive forces compared to zebrafish or killifish, this might suggest that the larger size of mammalian lenses requires a stronger mechanical integrity, for example, to withstand the mechanical stress due to accommodation by the ciliary body compared to piscine species that either move the whole lens to accommodate or, in the case of zebrafish, do not accommodate [23].

## Figures and Tables

**Figure 1 cells-10-02005-f001:**
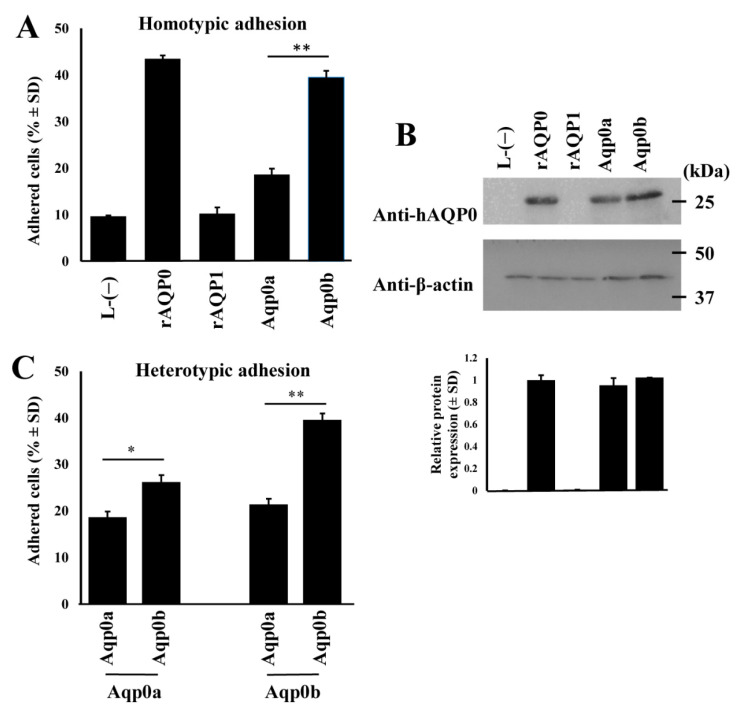
Zebrafish Aqp0b has adhesive properties. (**A**) Adherent cell percentages are shown for homotypic pairs of stable cell lines expressing positive control rat AQP0 (rAQP0), negative control rat AQP1 (rAQP1), zebrafish Aqp0a or Aqp0b, as well as cells transfected with an empty vector (L-(−)). Each lane represents *n* = 6–8. (**B**) Similar expression levels of rat AQP0 and zebrafish Aqp0 proteins shown by Western blotting using an anti-human C-terminus AQP0 antibody, and an anti-β-actin antibody as the loading control. Sample blot, top panel; quantification, bottom panel (*n* ≥ 3). (**C**) Heterotypic adhesion tested between zebrafish orthologs. Aqp0 plated first is indicated at the bottom. Statistical differences are shown as * having *p* < 0.05 or as ** having *p* < 0.01.

**Figure 2 cells-10-02005-f002:**
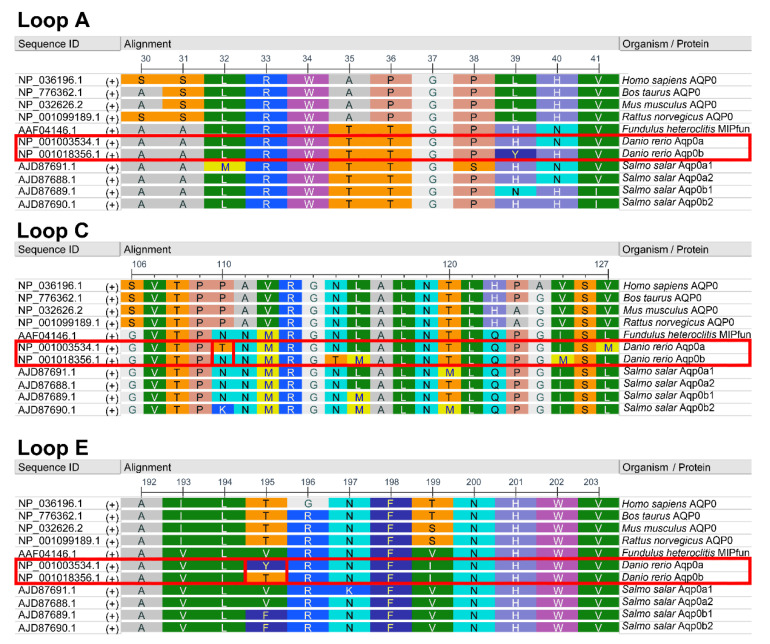
Zoo protein alignments of AQP0 extracellular loops. Amino acid alignment of extracellular loops A, C, and E of AQP0 across selected mammalian and piscine species. RasMol coloring. Human (*Homo sapiens*), bovine (*Bos taurus*), mouse (*Mus musculus*), and rat (*Rattus norvegicus*) AQP0 (also known as MIP) are shown. The fish Aqp0s shown include MIPfun of killifish (*Fundulus heteroclitus*) and duplicated Aqp0s for zebrafish (*Danio rerio*) and Atlantic salmon (*Salmo salar*). Two amino acids of interest indicated as boxed sites 110 and 195 were mutated in this study.

**Figure 3 cells-10-02005-f003:**
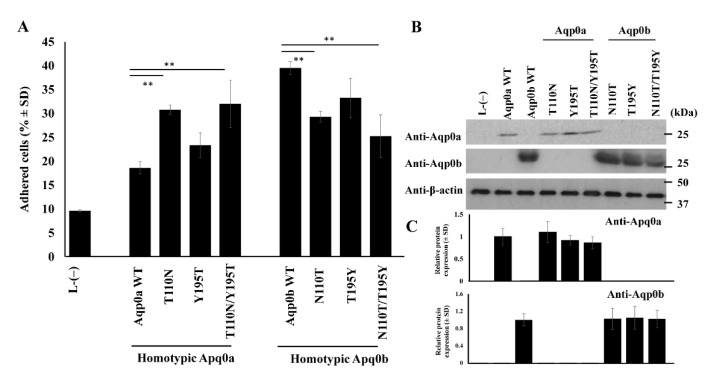
Mutational analysis reveals the importance of site 110 for adhesion of Aqp0b. (**A**) Adhesive properties were tested between homotypic pairs of stable cell lines expressing WT and mutant variants of Aqp0a and Aqp0b. Each lane represents *n* = 6–8. (**B**) Western blot analysis example showing equal expression of the protein between Aqp0 WT and mutant variants using zebrafish specific rabbit anti-Aqp0a and anti-Aqp0b sera quantified in (**C**) (*n* ≥ 3). Statistical differences are shown as ** having *p* < 0.01.

**Figure 4 cells-10-02005-f004:**
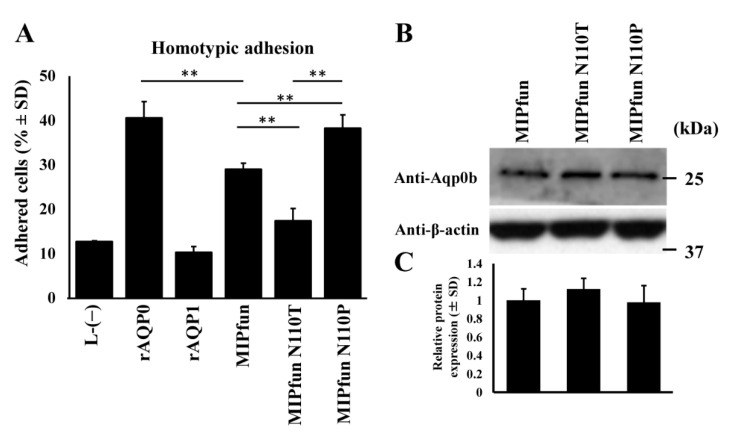
MIPfun’s adhesive properties are enhanced by an N110P mutation. (**A**) Adhesive properties were tested between homotypic pairs of stable cell lines expressing WT MIPfun and mutated variants. Each lane represents *n* = 6–8. (**B**) Western blot analysis example shows equal expression of the protein between WT and mutant variants using the rabbit anti-zebrafish Aqp0b sera quantified in (**C**) (*n* ≥ 3). Statistical differences are shown as ** having *p* < 0.01.

**Table 1 cells-10-02005-t001:** Summary of key C-loop amino acids (110–113) and self-adhesive properties of AQP0 from different species. Hs, *Homo sapiens*; Bt, *Bos taurus*; Ms, *Mus musculus;* Rt, *Rattus norvegicus;* MIPfun, *Fundulus heteroclitus*; Dr, *Danio rerio*; Ss, *Salmo salar*.

Species/Protein	C-Loop	Auto-Adhesive	Reference
HsAQP0 BtAQP0 MsAQP0 RtAQP0 MIPfun SsAqp0a1 SsAqp0a2 SsAqp0b1 SsAqp0b2 DrAqp0a DrAqp0b	PPAVR PPAVR PPAVR PPAVR PNNMR PNNMR PNNMR PNNMR PKNMR PTNMR PNNMR	Yes ? Yes Yes Yes Yes Yes Yes Yes No Yes	(Kumari and Varadaraj, 2009) (Varadaraj and Kumari, 2018) (Nakazawa et al., 2017) This study (Chauvigné et al., 2016) (Chauvigné et al., 2016) (Chauvigné et al., 2016) (Chauvigné et al., 2016) This study This study
HsAQP1 106–110	SSLTG	No	(Kumari and Varadaraj, 2009)
DrAqp08b	SSE	No	(Chauvigné et al., 2016)

## Data Availability

Upon request, we will make the data available to other researchers.

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
