# Peer review of "Differences in a Single Extracellular Residue Underlie Adhesive Functions of Two Zebrafish Aqp0s"

_cells, 2021, doi:10.3390/cells10082005_

Round 1

Reviewer 1 Report

In this manuscript, entitled “Differences in a Single Extracellular Residue Underlie Adhesive Functions of Two Zebrafish Aqp0s” the authors have assessed the adhesive properties of Aqp0a and Aqp0b in zebrafish using mouse fibroblast L-cells expressing pcDNA-Aqp0a, pcDNA-Aqp0b, or pcDNA-MIPfun. Using homotypic and heterotypic in vitro cell-to-cell adhesion assays they demonstrate that Aqp0b has strong auto-adhesive properties whereas Aqp0a does not. Furthermore, they report that the key residues differing in zebrafish Aqp0 are T110 and Y195 and demonstrate through point mutations followed by adhesion assays that the differences in adhesive properties appear to be due to residue 110, which is T110 in Aqp0a and N110 in Aqp0b. Finally, the authors show that mutating N110 in MIPfun to N110T also reduces adhesion similarly to N110 in Aqp0b. The manuscript contains a set of novel interesting data and the conclusions are supported by the experimental evidences. Nevertheless, in my opinion, the description of the applied methods as well as the presentation of the results need to be improved.

(1) Please correct spelling errors in lines 109 (Milliopore), line 111 (polylconal), 86 (MIPfin, this also needs correction in Figure 4B) and 202 (MIPFun).

(2) To improve the intelligibility, the authors might want to add “cells of each cell line” in line 86. Furthermore, can the authors be a little bit more precise about which antibodies have been used in which experiments and also give information on the secondary antibodies that were used in this study? Also, the identities with zebrafish Aqp0 given in the manuscript and suppl. Table 1 are quite low, have the authors considered performing an IP?

(3) The illustrations of Western Blot results lack information regarding kDa which should be included. The authors might also want to add the information of homotypic adhesion to Figure 1A and 4A. Also, since Figure 1C shows differences in heterotypic cell adhesion, the three bars on the left referring to controls ( L-(-), rAQP0 and rAQP1 ) are dispensable in my opinion as they are already shown in Figure 1A and not commented on neither in the text nor in the caption for Figure 1C. Instead, including dot blots of respective FACS analysis might improve the graphical representation of the results. Furthermore, in the caption of Suppl. Figure 1A the authors state that the „living cells“ were gated. Since only the forward and sideward scatter plot is displayed but no staining to discriminate living and dead cells, how do the authors conclude that the cells in their FSC/SSC gate are only living cells?

(4) The authors report that „MIPfun stably expressed in L-type mouse fibroblast cell lines exhibited homotypic adhesive properties ~21% (Figure 4A)” (lines 200-201). However, the respective bar in the bar graph of Figure 4 seems to indicate a higher percentage. Furthermore it is stated “Like, Aqp0b, MIPFun has N110, and mutating it to N110T reduced its adhesion slightly, though not statistically significant.” (lines 201-203) In contrast, in Figure 4A the difference between MIPfun and MIPfun N110T is illustrated as **, indicating a p-value <0.01, thus there is a discrepancy between the text and the illustration. Also, there is no statistical information regarding the difference between MIPfun and MIPfun N110P, neither in the text nor in the illustration.

Author Response

Answers to Reviewer #1

Thank you very much for reviewing our manuscript. As indicated in the responses that follow, we have taken all of these comments into consideration in the revised version.

For your convenience, our responses are written in red italics.

The manuscript contains a set of novel interesting data and the conclusions are supported by the experimental evidences. Nevertheless, in my opinion, the description of the applied methods as well as the presentation of the results need to be improved.

Thank you very much for your positive comments and constructive criticisms.

(1) Please correct spelling errors in lines 109 (Milliopore), line 111 (polylconal), 86 (MIPfin, this also needs correction in Figure 4B) and 202 (MIPFun).

They have been corrected.

(2) To improve the intelligibility, the authors might want to add “cells of each cell line” in line 86.

Added

Furthermore, can the authors be a little bit more precise about which antibodies have been used in which experiments and also give information on the secondary antibodies that were used in this study?

More information on specific antibodies used in each figure has been added to figure legends and can be found in greater detail in the methods (lines109-119). Secondary antibody used in this study was HRP-conjugated anti-rabbit antibodies (Cell Signaling Technology, Danver, MA, USA). This information has been added in the Methods (line 119).

Also, the identities with zebrafish Aqp0 given in the manuscript and suppl. Table 1 are quite low, have the authors considered performing an IP?

The purpose of using the Santa Cruz anti-human AQP0 antibody in figure 1B was to detect both zebrafish Aqp0a and Aqp0b with similar affinity. While we agree that the % identities of the antigenic peptide used to generate this antibody against zebrafish Aqp0a (61.36) and Aqp0b (54.55) are not high, they are nearly the same for both forms (Supplementary Table 1). Also, the Western blot as seen in Figure 1B clearly detects bands ~28kDA in cell lysates from lines expressing only Aqp0a or Aqp0b. These lines are absent in negative controls allowing us to detect and quantify the relative protein expression of both zebrafish Aqp0 orthologs. Therefore, in our opinion immunoprecipitation is superfluous, as the antibody clearly detects the two Aqp0 orthologs from cell lysates.

(3) The illustrations of Western Blot results lack information regarding kDa which should be included.

kDa have been added to all Western blots.

The authors might also want to add the information of homotypic adhesion to Figure 1A and 4A.

We have added titles to figures to clarify that Figure 1A and Figure 4A represent homotypic adhesion, and Figure 1C represents heterotypic adhesion assay.

Also, since Figure 1C shows differences in heterotypic cell adhesion, the three bars on the left referring to controls (L-(-), rAQP0 and rAQP1) are dispensable in my opinion as they are already shown in Figure 1A and not commented on neither in the text nor in the caption for Figure 1C.

This is a good point. We have taken out the negative controls from Figure 1C.

Instead, including dot blots of respective FACS analysis might improve the graphical representation of the results. Furthermore, in the caption of Suppl. Figure 1A the authors state that the „living cells“ were gated. Since only the forward and sideward scatter plot is displayed but no staining to discriminate living and dead cells, how do the authors conclude that the cells in their FSC/SSC gate are only living cells?

Thank you for picking up on this – instead of “living cells”, our text should state “single cells”, which has now been corrected in the figure legend of Supplementary Figure 1.

(4) The authors report that „MIPfun stably expressed in L-type mouse fibroblast cell lines exhibited homotypic adhesive properties ~21% (Figure 4A)” (lines 200-201). However, the respective bar in the bar graph of Figure 4 seems to indicate a higher percentage.

Text has been changed to reflect the 30% change.

Furthermore it is stated “Like, Aqp0b, MIPFun has N110, and mutating it to N110T reduced its adhesion slightly, though not statistically significant.” (lines 201-203) In contrast, in Figure 4A the difference between MIPfun and MIPfun N110T is illustrated as **, indicating a p-value <0.01, thus there is a discrepancy between the text and the illustration.

There is a significant decrease in adhesion in MIPfun N110T from WT MIPfun, and we have changed text to say “Like, Aqp0b, MIPfun has N110, and mutating it to N110T, as for Aqp0b, reduced its adhesive properties.”

Also, there is no statistical information regarding the difference between MIPfun and MIPfun N110P, neither in the text nor in the illustration.

We have added ** into Figure 4A to indicate statistically significance with a p-value <0.001. We have also added text in the discussion line 281 “This suggests that mammalian AQP0 may have stronger cell-to-cell adhesion than the piscine species, possibly to support mechanical integrity of the larger mammalian lens.”.

Reviewer 2 Report

As a reviewer, it was a joy reading this manuscript. I have no recollection of having dealt with a more complete and balanced study at the first round of reviews. The markings above speak for themselves. The only detail to hope for is presentation of n and p values in figures, figure legends or in results throughout the MS.

Author Response

As a reviewer, it was a joy reading this manuscript. I have no recollection of having dealt with a more complete and balanced study at the first round of reviews. The markings above speak for themselves. The only detail to hope for is presentation of n and p values in figures, figure legends or in results throughout the MS.

Thank you very much for reviewing our paper and for your kind words. All adhesion assays have n = 6-8 and Western blot assays have n of least 3. There were several p values missing from figure legends. We apologize for the oversight and have now ensured these are all present. Thank you for picking up on this. We have added these details to all figure legends.

Round 2

Reviewer 1 Report

The authors have carefully revised the manuscript and have replied appropriately to my critiques.

I have one minor critique left regarding the caption of Supplementary Figure 1. The authors now changed "gating of living cells" into "gating of single cells" which is also uncorrect, since a singlet gate is lacking and the depicted gate is applied to all measured cells without discerning between living, dead, singlets, doublets, etc. Thus, the authors might want to change the respective phrase into "gating of measured cells".

After this amendment, I consider this manuscript suitable for publication.

Author Response

The authors have carefully revised the manuscript and have replied appropriately to my critiques.

Thank you very much again for your constructive criticisms.

(1) I have one minor critique left regarding the caption of Supplementary Figure 1. The authors now changed "gating of living cells" into "gating of single cells" which is also uncorrect, since a singlet gate is lacking and the depicted gate is applied to all measured cells without discerning between living, dead, singlets, doublets, etc. Thus, the authors might want to change the respective phrase into "gating of measured cells".

Thank you for your comments. 

We have changed the legend of figure S1 from “living cells” to “gating of measured cells”.